# ARIA-MIDI: A DATASET OF PIANO MIDI FILES FOR SYMBOLIC MUSIC MODELING

**Louis Bradshaw, Simon Colton**
Queen Mary University of London
{l.b.bradshaw, s.colton}@qmul.ac.uk

## ABSTRACT

We introduce an extensive new dataset of MIDI files, created by transcribing audio recordings of piano performances into their constituent notes. The data pipeline we use is multi-stage, employing a language model to autonomously crawl and score audio recordings from the internet based on their metadata, followed by a stage of pruning and segmentation using an audio classifier. The resulting dataset contains over one million distinct MIDI files, comprising roughly 100,000 hours of transcribed audio. We provide an in-depth analysis of our techniques, offering statistical insights, and investigate the content by extracting metadata tags, which we also provide. Dataset available at https://github.com/loubbrad/aria-midi.

## 1 INTRODUCTION

Central to the success of deep learning as a paradigm has been the datasets used to train neural networks. With the rapid technical advancements and ever-increasing availability of computational power, music has become a popular target for deep learning research, and deep learning in turn has had a notable impact on the study and creation of musical works (Briot et al., 2019). The progress of music-oriented deep learning depends heavily on access to diverse, well-structured datasets. Music is inherently structured and can be represented computationally in a variety of forms (Wiggins, 2016). In this work, we focus on symbolic representations of music, such as MIDI (Musical Instrument Digital Interface), which are widely used for encoding, analyzing, and facilitating the generation of musical compositions by both humans and machines (Ji et al., 2023).

In fields outside of computational music, the significance of comprehensive datasets is well established. For example, in computer vision, the ImageNet dataset (Deng et al., 2009) catalyzed research for almost a decade, providing both high-quality training data and robust benchmarks. Similarly, datasets such as Common Crawl (2024), C4 (Raffel et al., 2020), and the Pile (Gao et al., 2020) have been instrumental in advancing the field of natural language processing. These resources have enabled the study of scaling-based approaches for language modeling, enhanced available techniques, and provided foundation models for researchers with restricted resources (Zhou et al., 2024).

The situation for music-oriented deep learning research is mixed. While numerous publicly available audio-based datasets exist (Gemmeke et al., 2017; Hawthorne et al., 2018; Thickstun et al., 2016; Bertin-Mahieux et al., 2011), symbolic datasets, which represent music in formats such as MIDI, ABC, or MusicXML, are comparatively lacking in both quality and quantity. The Lakh dataset (Raffel, 2016), comprising 176,581 MIDI files scraped from the internet, has been widely adopted in model training (Thickstun et al., 2023; Zeng et al., 2021) due to its scale. However, its files, predominantly created through software sequencing or digital score conversion, often lack the nuances of expressive human performances, and vary significantly in quality. In contrast, the MAESTRO dataset (Hawthorne et al., 2018) offers high-quality Disklavier MIDI recordings from professional pianists, capturing the subtleties of human interpretation. However, its size and focus on virtuosic classical piano performances limit its applicability across diverse musical genres and styles.

Underlying this is a common limitation: the manual transcription process creates bottlenecks for both the scale and quality. In recent years, researchers have turned to Automatic Music Transcription (AMT) (Benetos et al., 2019) to address these limitations, creating various large-scale symbolic datasets (Kong et al., 2020; Zhang et al., 2022; Edwards et al., 2023). Leading AMT techniques leverage neural networks to extract symbolic note-level information from audio (Sigtia et al., 2016;

Table 1: Comparison of various publicly available datasets of symbolic music.

| Dataset | # Files | # Hours | Genre | Source | Multi-track |
|---------|---------|---------|-------|--------|-------------|
| MAESTRO | 1,276 | 199 | Classical | Piano Competitions | No |
| Mutopia[1] | 1,862 | 69 | Mixed | Lilypond | Yes |
| PiJAMA | 2,777 | 223 | Jazz | AMT | No |
| GiantMIDI | 10,855 | 1,237 | Classical | AMT | No |
| ATEPP | 11,742 | 1,009 | Classical | AMT | No |
| Lakh[2] | 176,581 | 9,567 | Mixed | Web-scrape | Yes |
| Aria-MIDI[3] | 1,186,253 | 100,629 | Mixed | AMT | No |

[1] https://www.mutopiaproject.org
[2] Full dataset size, including corrupted files. The commonly used *matched* subset contains 45,129 files.
[3] Reduces to 800,973 files and 66,650 hours after compositional deduplication described in Section 4.

Hawthorne et al., 2017; Kong et al., 2021), theoretically enabling symbolic datasets to match the scale found in other modalities. Nevertheless, several challenges persist:

**Transcription Quality.** Some musical formats, like solo-piano recordings, translate more accurately to MIDI than others. Additionally, training neural-AMT models relies on a small number of specific high-quality datasets of aligned audio-MIDI, e.g., MAESTRO for solo-piano. This limitation restricts use cases outside the distribution of the training data, leading to degraded transcriptions of recordings in different genres or with audio artifacts (Marták et al., 2024; Edwards et al., 2024).

**Pre-processing.** A dichotomy between quality and scale still presents itself. Current methods employed for audio curation and pre-processing (e.g., selection, pruning, and segmentation) are insufficient when applied to noisy and diverse audio corpora without human oversight. This is partially due to a lack of training data precisely labeled for the nuances of this application. Datasets maintaining high-quality standards have utilized a stage of machine-guided manual human verification to remove falsely identified audio (Zhang et al., 2022; Edwards et al., 2023), an approach that does not scale well.

In this work, we address these challenges, developing techniques for precise curation, pre-processing, and metadata attribution of publicly available piano recordings. We demonstrate that with strategic modifications to the data pipeline, AMT-based approaches can be effectively scaled, resulting in a large, high-quality dataset of piano transcriptions. To this end, we leverage Aria-AMT (Bradshaw et al.), a piano transcription model designed to handle diverse timbres and recording qualities. Although this model is important to our process, our primary focus is on the techniques we develop and the analysis of the resulting dataset's content and quality.

## 1.1 CONTRIBUTIONS OF THIS PAPER

More specifically, our contributions are as follows:

1. We introduce a scalable, language model-guided method for crawling and extracting metadata from specific categories of videos. We analyze the effectiveness of this approach in the context of publicly available piano recordings.

2. We outline a process for distilling an audio source-separation model to train a classifier capable of accurately identifying and segmenting diverse real-world piano recordings, which we open-source[1]. This enabled an 8-fold improvement in identification of non-piano audio without human supervision when compared to previous work.

3. Using an existing piano-transcription model, we provide a new MIDI dataset of piano transcriptions, Aria-MIDI, one of the largest and cleanest to date.

We hope the dataset released alongside this work has a positive impact on the MIR research community. We foresee several potential areas where it may accelerate research. Firstly, pretrained generative models have had a large impact on the textual and visual domains (Zhou et al., 2024).

[1] https://github.com/loubbrad/aria-cl

These models rely on datasets typically in terabytes. Comprising approximately 20 gigabytes of MIDI files, Aria-MIDI isn't on this scale; however, it may still be useful for research into pretrained music models. Secondly, we are releasing accurate compositional metadata for each file, as well as piano audio classifier scores, which due to our training methods can act as a proxy for recording quality. This information is valuable for many MIR tasks (Choi et al., 2017), as well as for making clean and *compositionally deduplicated* subsets.

## 1.2 RELATED WORK

The use of neural networks for automatic music transcription has its roots in the seminal work of Sigtia et al. (2015; 2016). This was followed by various works experimenting with different approaches and neural architectures (Hawthorne et al., 2017; 2021; Yan et al., 2021; Toyama et al., 2023). The high-resolution piano transcription model introduced in Kong et al. (2021), trained using the MAESTRO (Hawthorne et al., 2018) and MAPS (Emiya et al., 2010) datasets, became the de facto benchmark for accuracy. More recently, AMT research has extended to other instruments (Riley et al., 2024) and multi-track transcription (Gardner et al., 2021; Chang et al., 2024), where it has seen success.

There are three predominant publicly available datasets of piano transcriptions, all of which utilized the transcription model introduced in Kong et al. (2021). GiantMIDI (Kong et al., 2020) was the first, comprising transcriptions of piano recordings matching names of musical works taken from the IMSLP (2006). From 143,701 initial recordings, 10,855 were identified by a model trained to detect solo-piano recordings. The ATEPP dataset (Zhang et al., 2022) took a different approach than GiantMIDI, focusing on repeat performances of standard classical piano repertoire, and using text-based techniques to determine opus and piece numbers. PiJAMA (Edwards et al., 2023), a dataset of jazz piano transcriptions, spans 120 different pianists across 244 recorded albums. Recordings were curated by matching tracks from albums performed by a manually curated list of pianists.

All three datasets utilized YouTube to match musical metadata with audio recordings and employed audio-based classifiers, trained using MAESTRO and AudioSet, to identify piano recordings. For ATEPP and PiJAMA, these classifiers were also used to remove applause and speech. The level of human intervention varied across datasets: GiantMIDI relied solely on automated processes, while ATEPP and PiJAMA incorporated manual checks. Table 1 presents a comparison of these datasets in context.

## 2 METHODOLOGY

In this section, we describe the methodology used to compile our dataset of MIDI files. Our approach consists of three distinct stages. First, we assemble a comparatively large corpus of candidate piano recordings using low-overhead, text-based methods. Next, we employ audio-based techniques to refine our initial corpus through pruning and pre-processing. Finally, we conduct a computationally intensive stage of transcription and metadata extraction. This multi-stage approach allows us to efficiently process a large volume of data while ensuring high-quality data in our final dataset.

## 2.1 CRAWLING

A common theme in previous work has been to compile candidate recordings by first obtaining a corpus of metadata (e.g., composers, performers, album titles) via various means, and then using the APIs provided by Spotify[2] and YouTube[3] to match piece titles to corresponding videos on YouTube. We take a different approach: Our method begins with a small collection of manually curated seed videos and uses YouTube's API to crawl related content. The crawling priority is determined by a language model, which performs two tasks: 1) parsing the title and description of each video, and 2) scoring the likelihood of the video containing solo-piano content on a scale of 1-5. More specifically, starting from fifty solo-piano seed videos which span a variety of genres and styles, we follow a two-step procedure which we cycle through repeatedly:

---

[2]https://developer.spotify.com/documentation/web-api
[3]https://developers.google.com/youtube/v3

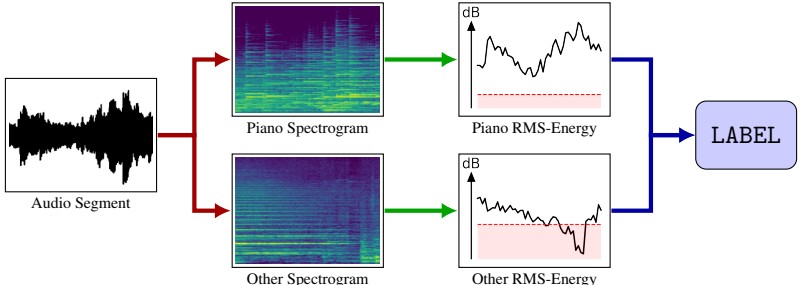

Figure 1: A visual representation of the pseudo-labeling process applied to a five-second excerpt from a piano concerto. As the non-piano component has a contiguous region above the energy threshold (dotted red line), the full audio segment is labeled as non-piano.

1. For each unscored video, we provide a system prompt to a language model along with the video's YouTube title and description. The model is tasked with assigning a score from one to five, indicating the likelihood that the video contains a solo-piano performance.

2. In order of priority determined by the score, we use the YouTube API to fetch related video URLs, titles, and descriptions.

By taking advantage of the related videos endpoint of the YouTube API, we outsource the majority of the crawling process to YouTube's own recommendation algorithm. We used the 70B parameter version of Llama 3.1 (Dubey et al., 2024) for the language model, observing that smaller models made obvious mistakes more frequently. The system prompt we used can be found in Appendix A.1. Overall, we found this process to be effective. Although initially this procedure tended to overrepresent recordings of well-known classical pieces, as these became less available later in the process, piano recordings representing a diverse set of musical styles and genres were crawled.

## 2.2 AUDIO CLASSIFICATION AND SEGMENTATION

As we analyze in-depth in Section 3, relying solely on the score attributed to each recording during the crawling process results in an unacceptably high rate of misclassifications. Following other work (Kong et al., 2020; Zhang et al., 2022), we address this by using an audio classification model (Dieleman and Schrauwen, 2014) in the next stage of our pipeline. We identified the following problematic situations which we aimed to mitigate with an audio classifier:

- Misclassifications due to logical mistakes by the language model or misleading/ambiguous YouTube data: The classifier identifies and removes such recordings while allowing us to retain those with some ambiguity, which would otherwise have to be pruned.

- Undesirable acoustic qualities in positively classified recordings: Despite positive classification by the language model, recordings can be inappropriate for transcription for a variety of reasons including incorrect instrumentation (e.g., harpsichord, organ, electric piano), low audio quality, or the presence of additional instruments. To mitigate this, we include representative examples from these categories in the training data for our classifier.

- Non-piano content in high-quality piano recordings: Many otherwise high-quality piano recordings contain segments of non-piano content, such as applause, commentary, or extended periods of silence. Using the algorithm we describe in Section 2.2.2, we adapt the audio classifier to segment recordings into contiguous regions of solo-piano performance, removing unwanted content.

A primary concern when building an audio classifier is the quality, diversity, and accuracy of the labels used for training. Initial investigations revealed that relying on well-known datasets such as MAESTRO (Hawthorne et al., 2018) and AudioSet (Gemmeke et al., 2017) was insufficient, as we display in Table 4. In an effort to achieve classification accuracy approximating human labels, we curated a mixed training dataset, representative of our corpus of crawled recordings. We used a novel approach, leveraging an audio source-separation model to accurately generate pseudo-labels for the unlabeled parts of the training corpus.

Table 2: Overview of the supervised and pseudo-labeled audio corpora used to train the piano audio classifier. *Prop.* denotes the proportion of audio in each component labeled as solo piano. Notably, when pseudo-labeled using source separation, the raw audio from which the GiantMIDI dataset was transcribed contains 87.35% solo piano labels.

| Component | Length (h) | Weight (%) | Pseudo lab. | Prop. (%) | $dB_{min}$ | $l_{min}$ (s) |
|---|---|---|---|---|---|---|
| GiantMIDI | 1040 | 43.24 | True | 87.35 | -25dB | 1.5 |
| Score-4 | 676 | 28.11 | True | 44.81 | -22dB | 1.5 |
| MAESTRO | 198 | 8.23 | False | 99.62 | N/A | N/A |
| Synthetic Data | 143 | 5.95 | False | 98.34 | N/A | N/A |
| Jazz Trio Database | 139 | 5.78 | True | 3.61 | -28dB | 1.0 |
| Piano Other | 75 | 3.12 | False | 99.22 | N/A | N/A |
| Non-Piano Other | 71 | 2.95 | False | 0.00 | N/A | N/A |
| Symphonies | 40 | 1.66 | False | 0.00 | N/A | N/A |
| Piano Concertos | 23 | 0.96 | True | 14.98 | -28dB | 1.0 |
| Total | 2405 | 100.0 | | | | |

Given an audio file, we apply the MVSep Piano source-separation model (Uhlich et al., 2024; Fabbro, 2024; Solovyev et al., 2023) to decompose it into its constituent parts, isolating the `piano` component from the remaining audio (`other`). For each five-second clip, we compute separate spectrograms for both the `piano` and `other` components after resampling them to 22,050 Hz. Each spectrogram is computed with 2048 frequency bins, a frame length of 2048, and a hop length of 512. The RMS energy is then computed for each frame and converted to the dBFS scale (Zölzer, 2022). Given parameters $dB_{min}$ and $l_{min}$, we label a five-second clip as *non-piano* if the `other` component contains a contiguous region longer than $l_{min}$ at an energy level exceeding $dB_{min}$. We also label a clip as non-piano if the `piano` component has a contiguous silent region (below -20 dB) lasting more than four seconds. In all other cases, the clip is labeled as *piano*. Figure 1 illustrates this process.

We applied this labeling procedure to various collections of publicly available audio files, displayed in Table 2, the main constituent being 10,000 YouTube videos from our corpus which were assigned a score of four by the language model. We also used the GiantMIDI audio files, the Jazz Trio Database (Cheston et al., 2024), and smaller collections of piano and non-piano recordings which we curated manually. Including the pseudo-labeled audio allows us to distill (Hinton et al., 2015) the source-separation model, bypassing the high computational cost associated with applying source separation to our entire inference corpus, which we estimate at about 5,000 A100 hours[4].

### 2.2.1 TRAINING

For our solo-piano classifier, we chose a CNN-based architecture (LeCun et al., 2015) with five convolutional layers followed by two dense layers and a single output neuron. The input to the classifier consists of mel-spectrograms calculated from five-second audio clips. We used a sample rate of 22,050 Hz, 2048 spectrogram frequency bins, 256 mel bins, and a hop length of 220 (corresponding to 10ms hops). We trained the model for ten epochs using the AdamW optimizer (Loshchilov and Hutter, 2019) with $\beta_1, \beta_2 = 0.9, 0.95$, $\epsilon = $ 1e-6 and an L2 weight decay of 0.01. A linear learning rate scheduler was used, decaying to 10% of the initial learning rate after a warmup over the first 500 optimizer steps.

One consequence of training with pseudo-labels obtained using source separation was having to use relatively sensitive energy thresholds in order to correctly label training examples with a quiet but notable non-piano component. These thresholds occasionally result in incorrect training labels for solo-piano recordings with significant but acceptable background audio artifacts like noise, distortion, and reverb. To mitigate this, we trained with the corresponding audio augmentation in approximately 10% of batches, as well as randomly applying pitch shifting and bandpass filters. We also included labeled examples, representative of such piano recordings, as part of our training data.

---

[4]In comparison, classification of 100,000 hours of audio using our model only took 20 A100 hours, I/O being the main bottle-neck.

### 2.2.2 INFERENCE

As well as per-file classification, we also use our classification model to segment audio recordings into their standalone components of contiguous piano performance. To do this, we employ a sliding-window based technique adapted from standard approaches (Keogh et al., 2004), aimed at accurately removing non-piano content whilst being robust to short-lived classification mistakes.

Given an audio recording, we score each five-second interval, sampled with a one-second stride, by passing the inputs through our model. We classify a region $(n, m + 5)$, $m \geq n + d$, as non-piano if and only if all segments starting between $n$ and $m$ are scored below $\lambda$. The parameters $d$ and $\lambda$ control the sensitivity and minimum length of non-piano segments, which we set to 3 and 0.5 respectively. After excluding all non-piano segments, we classify the remaining contiguous segments as piano if they are longer than 45 seconds. Finally, we discard piano segments with an average score lower than 0.7. Our choice of algorithm and hyperparameters was motivated to reduce the chance of a solo-piano segment being prematurely interrupted due to instability in scoring. Since our classifier designates intervals that are mostly silent as non-piano, this approach also separates segments of piano performances that are separated by at least $d + 5$ seconds of silence. In Section 3, we investigate the accuracy of both classification and segmentation of our proposed approach.

### 2.3 TRANSCRIPTION

We used a Whisper-based model, Aria-AMT[5] (Bradshaw et al.), to transcribe the segmented audio recordings into MIDI files. This choice was informed by the model's robustness in transcribing audio from a diverse set of recording environments, compared to models used in previous work (Kong et al., 2020; Zhang et al., 2022) (See Appendix A.3). Transcription of the 100,629 hours of audio took 765 hours using an NVIDIA H100 GPU with a batch size of 128, representing an inference speed of roughly 131x real-time, transcribing approximately 2327 notes per second.

### 2.4 METADATA EXTRACTION

Access to per-file metadata labels enables flexible dataset splits for generative and MIR tasks. A central concern of ours was *entity resolution* (ER) (Christen, 2011), i.e., identifying the compositional source of each recording and using this information to mitigate overrepresentation of popular pieces[6].

Inspired by our crawling methodology, we applied a similar approach to metadata extraction. Using Llama 3.1 (70B), we processed YouTube titles and descriptions for files that passed language model and audio detection filters. The prompt (see Appendix A.2) extracted composer, opus numbers (e.g., Op., BWV, K., D.), piece identifiers, performer, genre, and form labels when such information was present in the text.. These metadata labels facilitate control over *compositional duplication* and provide supervised labels for MIR. We analyze their accuracy and distribution in Sections 3 and 4.

## 3 METHODOLOGICAL ANALYSIS

In this section, we evaluate the effectiveness of the components in our data pipeline. Where applicable, we compare our methods to those used in previous work, in particular the GiantMIDI, ATEPP, and PiJAMA datasets. For baselines and to determine ground truth, we relied on human labels obtained from two musically trained pianists familiar with popular classical and jazz repertoire.

**Language Model Classification.** We first analyze the ability of a language model to correctly classify a video as solo-piano according to its YouTube title and description. In our experiment we chose a random sample of 250 videos from those crawled, and calculated the accuracy of the labels provided by different language models, judged relative to the ground truth in the audio content. We also asked human participants to label the videos according to the same prompt given to the language models. Classification precision, recall, and F1 scores can be seen in Table 3. In comparison to human-derived labels, language models perform well at this text-based classification task. We attribute this to the depth of knowledge of different composers, performances, and pieces, which the

---

[5]https://github.com/eleutherai/aria-amt
[6]For example, *moonlight* appears in 6,819 titles, likely referring to Beethoven's Moonlight Sonata.

Table 3: Classification precision, recall, and F1-scores for different models in the Llama 3.1 family, as well as human labels, across various score classification thresholds. Results suggest that Llama 3.1 70B with a score threshold of 4 provides a good balance between inference cost and accuracy.

| Model | Score ≥ 3 | | | Score ≥ 4 | | | Score ≥ 5 | | |
|---|---|---|---|---|---|---|---|---|---|
| | P (%) | R (%) | F1 (%) | P (%) | R (%) | F1 (%) | P (%) | R (%) | F1 (%) |
| Llama 3.1 8B | 73.08 | 81.43 | 77.03 | 82.26 | 72.86 | 77.27 | 86.36 | 27.14 | 41.30 |
| Llama 3.1 70B | 64.76 | 97.14 | 77.71 | 70.83 | **97.14** | 81.93 | 84.51 | 85.71 | 85.11 |
| Llama 3.1 405B | 77.01 | 95.71 | 85.35 | 80.49 | 94.29 | **86.84** | **94.44** | 72.86 | 82.26 |
| Human labels | 73.63 | 95.71 | 83.23 | 83.56 | 87.14 | 85.31 | 85.71 | 25.71 | 39.56 |

language models have access to. Despite this, there remains a clear discrepancy between the audio ground truth and the labels obtained from YouTube titles and descriptions alone.

**Audio Segmentation.** We evaluate the performance of our audio classification model in identifying and segmenting solo-piano content within audio recordings. For comparison, we used MVSep directly to obtain binary labels, applying the same inference procedure as described in Section 2.2.2. For ablation, we trained a model without the pseudo-labeled training data listed in Table 2. To mimic classifiers used for segmentation in other work, notably for the GiantMIDI, ATEPP, and PiJAMA datasets, we include various noise, applause, and speech from AudioSet (Gemmeke et al., 2017) as negative training examples for our ablation model.

For this analysis, a random sample of 250 audio recordings assigned language model scores greater than or equal to 3 was selected, excluding those used during training. To establish ground truth, our two participants were tasked with segmenting recordings into regions of solo-piano content and assigning files into one of three categories: Not solo-piano, solo-piano with significant audio artifacts, or solo-piano with good to pristine recording quality. Human-labeled segments were post-processed in accordance with our inference algorithm: Non-piano segments shorter than eight seconds were ignored, and a minimum length of 45 seconds was imposed on piano segments. Segmentation accuracy results can be seen in Table 4. While the ablation model achieves accurate segmentation accuracy, being less likely to interrupt piano segments by misclassifying occasional noisy periods of extreme piano audio as non-piano content, it mislabels non-piano audio as piano eight times more frequently than the proposed approach in absolute terms.

We next evaluated our model's classification performance. To assess this on a per-file basis, we imposed minimum thresholds on the average audio score for predicted piano segments, negatively classifying files with no predicted piano segments after filtering. Table 5 reports the accuracy of this approach in identifying non-piano recordings as well as solo-piano recordings with significant audio artifacts in our evaluation dataset. Additionally, we analyzed the audio files that constitute Giant-MIDI, ATEPP, and PiJAMA. Our human participants manually categorized files in these datasets that fell below an empirically determined average score threshold of 0.7, indicating issues with recording quality or content. The resulting distributions of these categorizations are shown in Figure 2.

In both tasks, our approach performs well. For segmentation with λ=0.5, we achieve a 96.38% overlap with the ground truth for high-quality piano recordings, while removing 98.83% of non-piano audio across the evaluation corpus. When applying a segment average-score threshold of T=0.7, we remove 100% of non-piano files on a per-file basis while retaining 95.28% of high-quality piano files. Furthermore, increasing T to 0.9 removes most low-quality piano audio files, enabling us to curate a clean dataset split which we also provide.

**Metadata extraction.** To evaluate our approach to metadata extraction, we selected a random sample of 200 files and cross-referenced the metadata labels assigned by the language model with the corresponding YouTube titles and descriptions. For each file and metadata category, we manually checked for incorrect labels (e.g., misattributions) and missing labels where relevant information was present in the raw text. This process provides accuracy estimates for the metadata labels in the final dataset. Notably, the language model sometimes generated accurate labels that were not present in the raw text. The results of this evaluation are shown in Table 6.

Table 4: Segmentation accuracy and overlap ratios for different techniques and hyperparameters. We consider a predicted segment correct if its beginning and end match the reference within tolerances of ±2 seconds and ±5 seconds, respectively. Each reference segment is matched to at most one predicted segment. Overlap ratios are calculated separately for piano and non-piano audio, each as the ratio of the duration of correctly identified audio to the total duration of the respective ground truth audio type. $dB_{min}$ and $\lambda$ denote the sensitivity to non-piano content as described in Section 2.2.

| | Segmentation Accuracy | | | Segment Overlap | |
|---|---|---|---|---|---|
| **Technique** | **P (%)** | **R (%)** | **F1 (%)** | **Piano (%)** | **Non-Piano (%)** |
| *Full corpus* | | | | | |
| MVSep, $dB_{min}$=-22dB | 65.62 | 70.47 | 67.96 | 91.96 | 97.22 |
| MVSep, $dB_{min}$=-25dB | 58.28 | 63.76 | 60.90 | 88.74 | 98.18 |
| MVSep, $dB_{min}$=-28dB | 49.38 | 53.69 | 51.45 | 82.23 | 98.66 |
| Proposed, $\lambda$=0.5 | **71.97** | 75.84 | 73.86 | 94.22 | 98.83 |
| Proposed, $\lambda$=0.6 | 70.70 | 74.50 | 72.55 | 92.67 | 98.89 |
| Proposed, $\lambda$=0.7 | 68.39 | 71.14 | 69.74 | 91.04 | **99.10** |
| Ablation, $\lambda$=0.5 | 71.18 | **81.21** | **75.86** | **97.05** | 91.10 |
| *All solo-piano recordings* | | | | | |
| MVSep, $dB_{min}$=-22dB | 68.18 | 70.47 | 69.31 | 91.96 | 89.50 |
| MVSep, $dB_{min}$=-25dB | 59.75 | 63.76 | 61.69 | 88.74 | 93.43 |
| MVSep, $dB_{min}$=-28dB | 50.00 | 53.69 | 51.78 | 82.23 | 94.12 |
| Proposed, $\lambda$=0.5 | 72.44 | 75.84 | 74.10 | 94.22 | 92.73 |
| Proposed, $\lambda$=0.6 | 71.15 | 74.50 | 72.79 | 92.67 | 93.18 |
| Proposed, $\lambda$=0.7 | 68.83 | 71.14 | 69.97 | 91.04 | **94.66** |
| Ablation, $\lambda$=0.5 | **77.56** | **81.21** | **79.34** | 97.05 | 73.33 |
| *Quality solo-piano recordings* | | | | | |
| MVSep, $dB_{min}$=-22dB | 70.00 | 75.97 | 72.86 | 94.71 | 85.13 |
| MVSep, $dB_{min}$=-25dB | 61.64 | 69.77 | 65.45 | 92.06 | 87.04 |
| MVSep, $dB_{min}$=-28dB | 50.33 | 58.91 | 54.29 | 87.41 | 88.22 |
| Proposed, $\lambda$=0.5 | 75.36 | 80.62 | 77.90 | 96.38 | 84.80 |
| Proposed, $\lambda$=0.6 | 73.91 | 79.07 | 76.40 | 95.33 | 86.13 |
| Proposed, $\lambda$=0.7 | 71.53 | 75.97 | 73.68 | 93.92 | **88.29** |
| Ablation, $\lambda$=0.5 | **82.09** | **85.27** | **83.65** | **97.15** | 83.13 |

Table 5: Classification performance for different segment average score thresholds, evaluated on our human-labeled dataset. Segments were calculated using $\lambda$=0.5. *All solo-piano* refers to the classification performance for identifying files containing solo piano segments, regardless of audio artifacts. *Quality solo-piano* refers to identifying only recordings with good to pristine audio quality. FP = False Positives.

| | All solo-piano | | | | Quality solo-piano | | | |
|---|---|---|---|---|---|---|---|---|
| **Threshold** | **P (%)** | **R (%)** | **F1 (%)** | **FP** | **P (%)** | **R (%)** | **F1 (%)** | **FP** |
| $T \geq 0.50$ | 99.28 | 93.24 | 96.17 | 1 | 88.49 | 96.85 | 92.48 | 16 |
| $T \geq 0.60$ | 99.27 | 91.89 | 95.44 | 1 | 89.05 | 96.06 | 92.42 | 15 |
| $T \geq 0.70$ | 100.00 | 89.86 | 94.66 | 0 | 90.98 | 95.28 | 93.08 | 12 |
| $T \geq 0.80$ | 100.00 | 85.14 | 91.97 | 0 | 90.48 | 89.76 | 90.12 | 12 |
| $T \geq 0.90$ | 100.00 | 75.68 | 86.15 | 0 | 95.54 | 84.25 | 89.54 | 5 |
| $T \geq 0.95$ | 100.00 | 61.49 | 76.15 | 0 | 97.80 | 70.08 | 81.65 | 2 |

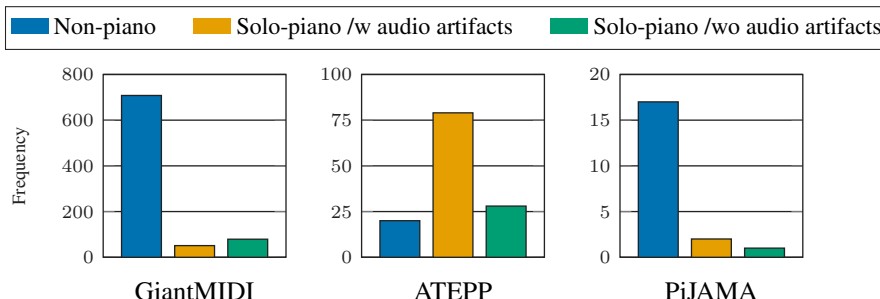

Figure 2: Distribution of files with average audio classifier scores $\leq 0.7$ across datasets. Files were manually categorized. The differences in distribution between datasets may be attributed to human-in-the-loop pruning in ATEPP and PiJAMA, which is absent in GiantMIDI..

Table 6: Analysis of metadata presence and accuracy across different attributes. For each attribute, presence indicates the percentage of files with assigned metadata, accuracy shows the percentage of correct labels among present metadata, and missed labels represents the percentage of files where metadata was omitted despite being inferrable from YouTube titles and descriptions. Accuracy was verified following the criteria specified in the system prompt (see Appendix A.2).

| Attribute | Presence (%) | Accuracy (%) | Missed Labels (%) |
|---|---|---|---|
| Composer | 71.0 | 99.3 | 2.7 |
| Performer | 62.0 | 99.2 | 0.8 |
| Opus Number | 32.0 | 100.0 | 1.5 |
| Piece Number | 22.0 | 93.2 | 4.3 |
| Key Signature | 23.0 | 97.8 | 0.0 |
| Genre | 86.5 | 94.2 | 0.6 |
| Music Period | 63.0 | 92.9 | 12.5 |

## 4   DATASET STATISTICS

In this section, we present statistics on our methodology and the resulting dataset of MIDI files. Overall, when executing our data pipeline, we collected YouTube data for 3,290,453 videos, from which we further processed 1,713,650 using our audio classifier. We then transcribed over 1 million audio segments into approximately 100,000 hours of transcribed solo-piano music. We present a breakdown of scores ascribed during each stage of processing in Figure 3. Taken together with the experiments in Section 3, we conclude that the techniques we have developed work well at scale.

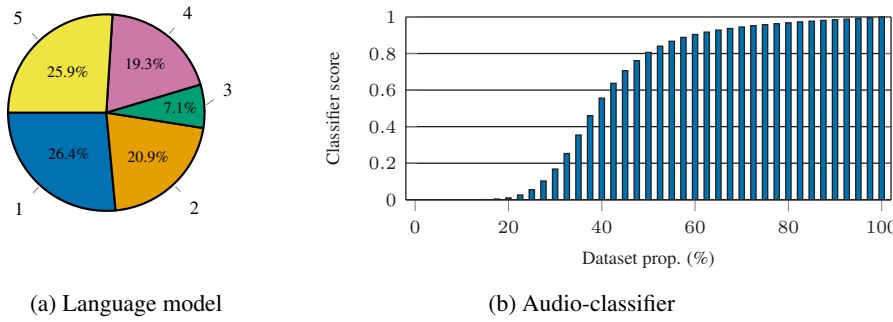

(a) Language model

(b) Audio-classifier

Figure 3: Score breakdowns for different components of our data pipeline. Figure (a) shows the distribution of language model scores during crawling. Figure (b) illustrates the cumulative distribution of audio classifier scores for recordings with a language model score of at least three.

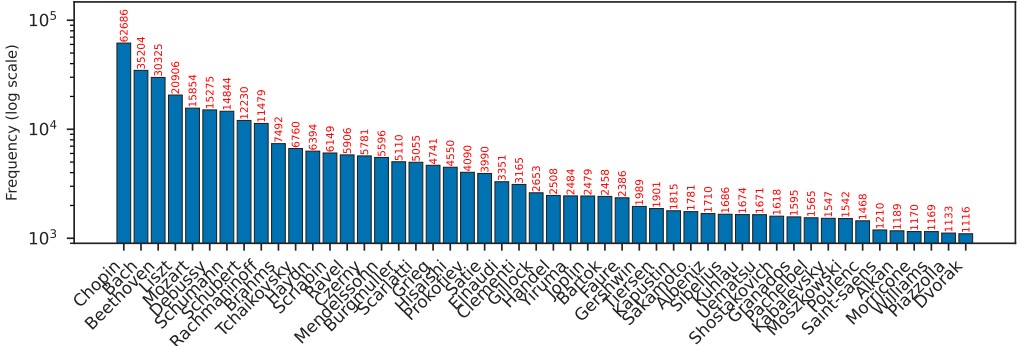

Figure 4: Number of transcriptions in Aria-MIDI attributed to the top 50 composers (log scale).

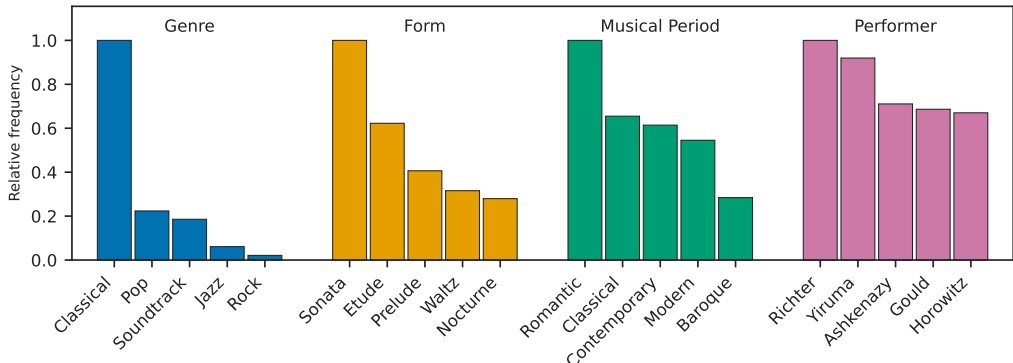

Figure 5: Relative frequency distribution of metadata across categories, normalized to the most frequent one. Common surnames in the performer category (e.g., Lee, Kim) are not displayed.

Moreover, extrapolating from the results in Table 5, we observe that the top-scoring 35,000 hours of MIDI files likely contain few transcriptions of non-solo-piano content.

To address compositional duplicates, we analyze metadata tags in three categories: composer, opus number, and piece number. To obtain a dataset split minimizing compositional duplicates within the text-based metadata constraints, we remove files that either match on all three tags (composer, opus number, and piece number) or match on both composer and opus number in cases where piece number tags are absent. For composers who appear more than 250 times across the dataset, we also prune all additional files that lack opus number and piece number tags. Overall, we identified 23,283 unique metadata triples, and after removing compositional duplicates using this procedure, 800,973 files remained. Figure 4 shows the frequency of performances of works by different composers in the complete dataset, illustrating their relative popularity.

Lastly, Figure 5 shows the distribution of metadata for other categories over the entire collection of MIDI files, without deduplication. Overwhelmingly, transcriptions of classical piano performances dominate; however, when accounting for the total size, many other genres are well represented.

## 5 CONCLUSION

We have introduced a new dataset of MIDI files, created by transcribing piano performances publicly accessible on the internet. In this paper, we provide an analysis of the components in our data pipeline and find them to be well-suited for our purposes. Going forward, we see several areas for future work: Primarily, extending our approach to other instruments such as guitar, as well as the multi-instrument case, could be approachable via variations of the source-separation-based approaches to audio pre-processing we have outlined. Secondly, further study into metadata attribution using language models, especially targeting improvements in compositional entity recognition.

ACKNOWLEDGMENTS

This research utilized the AI Industrial Convergence Cluster supported by the Ministry of Science and ICT of Korea, and Gwangju Metropolitan City. We also thank Roman Sol for making the MVSep piano source separation model available to us, and Alex Spangher for his assistance with audio processing.

COPYRIGHT DISCLAIMER

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

# A  APPENDIX

## A.1  CRAWLING SYSTEM PROMPT

```
Analyze the YouTube video title and description to determine if it's
    likely a solo piano performance. Consider the following:

1. Is the content music-related?
2. Are there explicit mentions of solo piano or pianist names?
3. Does it mention other instruments, vocalists, or non-musical
    elements?
4. Is it an educational video (tutorial, lesson) rather than a
    performance?
5. If a piece name is provided, is it typically for solo piano?

Pay special attention to these factors, which suggest the content is
    NOT a pure solo piano performance:

- Presence of other instruments or vocalists
- Educational content (lessons, tutorials)
- Non-piano keyboard instruments (e.g., organ, harpsichord)
- Significant narration or spoken content
- Orchestral accompaniment (e.g., piano concertos)
- Audio content beyond solo piano
- Repetitive tracks (e.g., loop videos)

The presence of any of these elements should generally result in a
    lower rating.

Assign a score from 0-5 where:

5 = Certainly a solo piano performance only
- Clear indication of a solo pianist performing
- No signs of additional instruments, vocals, or non-performance
    elements

4 = Very likely a solo piano performance, but not entirely certain
- Strong indications of solo piano, but some minor ambiguity
- No clear signs of additional elements, but not explicitly ruled out

3 = Possibly a solo piano performance, but with significant
    uncertainty
- Some indications of solo piano, but also hints of potential
    additional elements
- Could be a piano-focused piece with minimal additional content

2 = Likely includes elements other than solo piano
- Clear indications of additional instruments, educational content,
    or non-performance elements
- Still primarily piano-focused, but definitely not a pure solo
    performance

1 = Mostly not a solo piano performance
- Significant presence of other instruments, vocals, or non-musical
    content
- Piano may be present but is not the sole or main focus

0 = Definitely not a solo piano performance or not piano-related at
    all
- No indication of solo piano content
- Completely unrelated to piano performances

Examples:
```

```
49 "Chopin Nocturne Op. 9 No. 2 - Arthur Rubinstein" => 5
50 "The Art of Fugue - Glenn Gould (Piano)" => 5
51 "Bohemian Rhapsody - Piano Cover with Sheet Music" => 4
52 "Beethoven - Ode To Joy | VERY EASY Piano Tutorial" => 3
53 "Mozart Piano Concerto No. 21 - London Symphony Orchestra" => 1
54 "Top 10 Guitar Solos of All Time" => 0
55
56 Think step by step concisely, and then provide your score as a JSON
      string: {"score": X}
```

### A.2  METADATA EXTRACTION SYSTEM PROMPT

```
1 Analyze the YouTube video title and description provided within XML
      tags. If it contains information about a solo-piano performance,
      extract the following metadata and provide it as a JSON string:
2
3 - composer: Last name of the composer, if applicable (string, omit if
      not present or uncertain)
4 - opus: Opus number (e.g., Op., BWV, K., D.), if applicable (integer,
      omit if not present or uncertain)
5 - piece_number: Number or identifier within the opus, if applicable (
      integer, omit if not present or uncertain)
6 - genre: Primary genre of the piece (string: "classical", "jazz", "
      pop", "blues", "ragtime", "atonal", "rock", "soundtrack", "
      ambient", "folk", omit if uncertain)
7 - form: Musical form (e.g., "sonata", "etude", "improvisation", "
      fantasy", etc.) (string, omit if unknown or not applicable)
8 - performer: Last name of the pianist or performer, if known (string,
      omit if unknown or uncertain)
9 - key_signature: Key signature of the piece (string, omit if not
      mentioned or uncertain)
10 - difficulty: Estimated difficulty level (string: "beginner", "
      intermediate", "advanced", "virtuoso", omit if uncertain)
11 - music_period: Primary musical period (string: "classical", "
      romantic", "baroque", "impressionist", "contemporary", "modern",
      omit if uncertain)
12
13 Rules:
14 1. Omit keys and values entirely for fields not present, unknown, or
      uncertain. Do not include empty strings or placeholder values.
15 2. Be cautious not to include fields unless you are reasonably
      certain they are correct.
16 3. Provide opus and piece_number as integers only (e.g. don't include
      BWV, K., S., or Op.). Omit if not clearly a number or if zero.
17 4. In the case of well-known pieces (e.g., Moonlight Sonata,
      Fantaisie-Impromptu, etc.), add the opus and piece_number if you
      are certain, even if it is not in the raw text.
18 5. Provide form as a single word each, using very general and well-
      known terms.
19 6. Infer difficulty and period from context when possible, but omit
      if uncertain.
20 7. For all strings only provide a single word in lowercase ASCII.
21 8. For composer and performer, use only the last name. If unsure
      which name is the last name, omit the field.
22 9. Provide key_signatures using standard ASCII musical notation: Use
      'b' for flat, '#' for sharp, and 'm' for minor. Major keys should
      not have a suffix. Examples: 'c', 'f#m', 'bb'.
23 10. Only include opus and piece_number if the video is a complete
      performance of a single traditional opus number (Op., BWV, K., D
      .) and its movements/variations. Omit both fields for
      compilations, multiple works, ambiguous titles, or when using non
      -traditional/modern catalog numbers.
24 11. Don't confuse piece_number with other identifiers like sonata
      numbers (e.g., "Sonata No. 14") or separate opus numbers (e.g.,
```

```
        in "Op. 37-38", neither 37 nor 38 is a piece_number). Only use
        piece_number when  i t s  part of an opus and subordinate to it.

Examples:

1. Input:
<title>Chopin - Nocturne in E-flat major, Op. 9 No. 2 | Rousseau </
    title>
<description>Fr d ric Chopin's Nocturne in E-flat major, Op. 9, No.
     2. One of the most famous classical piano pieces from the
    Romantic era. Performed by Rousseau.
#chopin #nocturne #classical #piano</description>

Output:
{
  "composer": "chopin",
  "opus": 9,
  "piece_number": 2,
  "genre": "classical",
  "form": "nocturne",
  "performer": "rousseau",
  "key_signature": "eb",
  "difficulty": "advanced",
  "music_period": "romantic"
}

2. Input:
<title>Glenn Gould plays Bach Partita No.2 in C-minor (FULL)</title>
<description>1959 Studio recording DISCLAIMER: I do not own any
    material shown in this video. This is for entertainment purposes
     ONLY. Unlawful distribution of this material can result in bad
    stuff, apparently, SO DON'T DO IT!</description>

Output:
{
  "composer": "bach",
  "genre": "classical",
  "form": "partita",
  "performer": "gould",
  "key_signature": "cm",
  "difficulty": "advanced",
  "music_period": "baroque"
}

3. Input:
<title>Jazz Piano - Bill Evans - The Solo Sessions, Vol1 [ Full Album
     ]</title>
<description></description>

Output:
{
  "performer": "evans",
  "genre": "jazz",
  "music_period": "modern"
}

4. Input:
<title>Martha Argerich plays Beethoven Sonata No. 31, Op. 110</title>
<description>00:00 1. Moderato cantabile molto espressivo
06:12 2. Allegro molto
08:18 3. Adagio ma non troppo - Allegro ma non troppo
</description>

Output:
{
```

```
81    "composer": "beethoven",
82    "opus": "110",
83    "genre": "classical",
84    "form": "sonata",
85    "performer": "argerich",
86    "difficulty": "advanced",
87    "music_period": "classical"
88 }
89
90 Think step by step concisely, and then provide the metadata as a JSON
       string.
```

## A.3    TRANSCRIPTION ACCURACY

Table 7: Piano transcription note accuracy of the transcription model used for Aria-MIDI, evaluated on the MAESTRO (v3) and MAPS test sets. Results are calculated using the mir_eval library (Raffel et al., 2014) with default settings. We compare to the model introduced in Kong et al. (2021), which was used for GiantMIDI, ATEPP, and PiJAMA.

| | Note | | | Note /w offset | | | Note /w offset & vel | | |
|---|---|---|---|---|---|---|---|---|---|
| | P (%) | R (%) | F1 (%) | P (%) | R (%) | F1 (%) | P (%) | R (%) | F1 (%) |
| *Chosen model* | | | | | | | | | |
| MAESTRO | 98.86 | 96.45 | 97.63 | 91.63 | 89.42 | 90.50 | 90.56 | 88.39 | 89.45 |
| MAPS | 91.78 | 89.47 | 90.58 | - | - | - | - | - | - |
| *Kong et al. (2021)* | | | | | | | | | |
| MAESTRO | 98.82 | 95.53 | 96.82 | 86.51 | 84.21 | 85.33 | 84.97 | 82.72 | 83.82 |
| MAPS | 79.37 | 87.43 | 83.10 | - | - | - | - | - | - |

