# OpenReview forum: "Aria-MIDI: A Dataset of Piano MIDI Files for Symbolic Music Modeling"
_ICLR.cc/2025/Conference — ICLR 2025 Poster_

### Official Review · Reviewer_NngF · 2024-10-28

**Soundness:** 2
**Presentation:** 3
**Contribution:** 2
**Rating:** 3
**Confidence:** 4

**Summary:**

This paper proposes a large-scale piano MIDI dataset through an automatic pipeline, comprised of a metadata extraction module, a source-separation guided classifier, and a MIDI transcriber.

**Strengths:**

This paper proposes a large-scale dataset, and details the creation process associated with it. With a large-scale dataset like this, the field of piano symbolic music generation can vastly benefit.

**Weaknesses:**

This paper lacks novelty, as the main work is centered around the creation of this dataset. The main process of dataset creation can be described as a cascade of open-source models and tools (such as llama, MVSep); these tools are widely available, and their combination is not novel. I aim to see some sort of generative experiments to demonstrate the superiority of this dataset (especially the scaling law effect the large-corpus should bring) or at least any neural networks trained on this proposed dataset, as without these experiments, I have no way of evaluating the validity of the dataset. As experiments are largely missing, I believe this paper still needs more work.

**Questions:**

1. Could the authors clarify if the proposed pipeline would be open-sourced, and if so, under which license?
2. Can the authors provide baseline experiment results on generation using this dataset against previous datasets, for example, GiantPiano-MIDI?

**Details Of Ethics Concerns:**

The dataset proposed could contain copyrighted information.

---

### Official Review · Reviewer_u1GW · 2024-10-31

**Soundness:** 4
**Presentation:** 4
**Contribution:** 3
**Rating:** 8
**Confidence:** 4

**Summary:**

This article proposes a MIDI dataset obtained from automatic transcription and scrapping. The data can be used for training generative models and for musicological analysis. It uses an LLM-powered scrapping method and an automatic music transcription algorithm to find MIDI files for a diversity of piano pieces.

**Strengths:**

The dataset is in fact extensive, and the data can be used to train generative music models. Together with the metadata, this could lead to a diversity of works related to music generation.

Originality: the idea of a dataset is not novel; however, a dataset this large is novel.
Quality: the paper is well written and addresses all issues related to the problem
Clarity: The only issue I found is that the fact that the dataset contains solely piano pieces should be mentioned in the title.
Significance: Highly significant, yet for a niche.

**Weaknesses:**

The dataset is a great contribution, but it might be controversial to say that the dataset falls under "fair use". This is, first, because "fair use" is different in each country; also, nothing in the CC-BY-NC licence mentions the use of models derived from this data to generate new data, that is, although the files cannot be sold themselves, the models able to generate similar excerpts could be sold and used for commercial purposes.  In this sense, a CC-BY-NC-SA would be more adequate and more coherent with the "fair use" idea, or, alternatively, terms of licencing that prohibit usage for commercial models or models that might be used in commercial distributions (avoiding a situation in which a free model is released, but is used commercially)?

According to https://www.copyright.gov/fair-use/, using creative work is less likely to support a "fair use" claim; also, using a large amount of copyrighted work for derivative creation could play agains the "fair use" claim. Could the authors add a discussion on how the dataset addresses each of the requisites for fair use?

It would be interesting to have a professional opinion - and maybe even community guidelines in the future - for this matter, as the liability could be large; if that issue is solved, the dataset is amazing. Could the authors include a detailed legal analysis - maybe derived from this consultancy with a copyright expert - and include their findings in the paper, particularly addressing potential liabilities and jurisdictional differences (remember, we have a worldwide community, so these rules might be different in each country).

**Questions:**

- Where will the dataset be hosted, and how will free access be granted?
- Will original creators be acknowledged?

---

### Official Review · Reviewer_41nW · 2024-11-03

**Soundness:** 3
**Presentation:** 3
**Contribution:** 2
**Rating:** 6
**Confidence:** 3

**Summary:**

The paper provides a dataset of one million MIDI files of expressive piano performances, and describes the process by which the dataset was constructed: auto-browsing YouTube for likely piano performances, downloading, auto-verifying and segmenting the piano audio, then transcribing the piano audio to MIDI.

**Strengths:**

It's awesome that this dataset is orders of magnitude bigger than other similar datasets and is being made publicly available.

The process for constructing the dataset, filtering, segmenting, assigning metadata, transcribing, etc., and checking the accuracy of each of these steps seems painstaking and I very much appreciate the work that has gone into the dataset construction.

**Weaknesses:**

Not a weakness per se, but this paper really seems like it should be at ISMIR.

I think the main thing I would have wanted to see but didn't is an effort to perform non-exact deduping on the *audio* (or MIDI).  I wouldn't at all be surprised if the same recording has been uploaded to YouTube multiple times with different titles, descriptions, etc., possibly including audio alterations.  Ideally if this did happen, only one instance would appear in the dataset.

Somewhat related (and partially out of my own curiosity), what fraction of the MIDI files are just Moonlight Sonata?  In my experience crawling YouTube, an outrageously large fraction of piano recordings are amateur recordings of Moonlight Sonata, in the single digit percents.  It might be worth emphasizing that blindly training a generative model on the dataset might yield undesirable results due to some weird skews (though of course the provided metadata should be able to help construct an appropriate training set).

While including only the MIDI files in the dataset and not the original audio may avoid some copyright issues, you might also want to remove *pieces* of music that are under copyright.  Since you have ostensibly accurate metadata for each performance, it shouldn't be too difficult to remove these, or at least to provide a version of the dataset that doesn't include these pieces.  Based on the statistics given in the paper, it seems likely that the *vast* majority of the pieces in the dataset are not under copyright and so this shouldn't reduce the size much.

The paper doesn't mention whether or not the transcribed performances include sustain pedal; some transcription models attempt to label sustain events and others don't.  Depending on the intended purpose of the dataset, it may be important to clarify.

**Questions:**

I do wonder why this wasn't submitted to ISMIR, as that community seems more suitable for this work.

---

### Official Review · Reviewer_VtYa · 2024-11-03

**Soundness:** 2
**Presentation:** 3
**Contribution:** 2
**Rating:** 3
**Confidence:** 4

**Summary:**

This paper introduces a dataset of over one million MIDI-files containing piano music. The files are obtained by transcribing music recordings from YouTube using an existing AMT model. As part of the dataset creation pipeline, YouTube is repeatedly crawled and a language model is used to score candidate video descriptions. Furthermore, audio recordings are segmented into piano/non-piano segments using a classification model distilled from an existing source separation model. Finally, dataset statistics are provided.

**Strengths:**

The novel dataset provided in this paper is a welcome contribution, especially given its very large size. The steps of the dataset preparation pipeline make sense to me and the paper provides several interesting insights and comparisons, e.g., on the effectiveness of different LLMs for pre-filtering videos or numbers on piano/non-piano classification and segmentation accuracies.

**Weaknesses:**

Overall, I am uncertain whether the dataset provided is a contribution significant enough to warrant inclusion at ICLR. This is especially because the dataset preparation pipeline consists in large parts of plugging together existing systems in a straightforward way (notably, the LLMs and AMT model). I am also skeptical about some of the evaluations provided and claims made in the paper (see "Questions" below).
This leads to my recommendation to reject the paper.

**Questions:**

Major questions/comments:
- Even though the provided MIDIs are the main contribution of the paper, the accuracy of the transcription model on the music segments that are ultimately transcribed by it is never evaluated. As such, it is unclear how clean the provided data actually is.
- Following up on the previous question, how would the transcription model have fared for piano/non-piano classification and segmentation?
- In lines 374ff, the accuracy of almost 98% for metadata extraction seems very, very high to me. Does this mean for 98% of songs, every single label (opus number, genre, performer...) is correct? Was this checked against the actual audio content? After all, a lot of YouTube videos are missing information about the exact piece being played or the performer in their description. How would a video that simply has the title "Favorite Chopin Prelude" (with no further information on which prelude exactly or which performer etc.) be evaluated in this context?
- In Table 4, it seems like the "Ablation" (without the pseudo-labeled training data) works much better than the "Proposed", which calls into question the use of this data and the proposed distillation of MVSep. Why would one still want to use the proposed approach?

Minor questions/comments:
- Line 44: It is claimed that the files in the Lakh dataset are primarily transcoded from MusicXML. I don't think that's true (are they not rather collected from various MIDI repositories on the internet?). Do you have a source for this claim?
- Table 1: The comparison in this table is misleading since the 1 million MIDIs in Aria-MIDI contain a lot of duplicates, whereas for example Mutopia and GiantMIDI contain unique compositions. Please clarify this in the table.
- Line 71ff notes "Instrumentation" as a persisting challenge in AMT and existing MIDI datasets. Line 84 claims "we address these challenges". However, the provided dataset contains only piano music, so this claim seems wrong to me.
- Line 174: It is stated that "piano recordings representing diverse set of musical styles and genres were crawled thoroughly." However, the final dataset overwhelmingly contains classical music. Were the other styles and genres filtered out in subsequent steps or is this statement here wrong?
- In Table 3: How can the Recall for Score >= 4 be higher than for Score >= 3 for the 70B model? The recall should decrease monotonically when changing the score threshold.

Additional comments:
- Line 28f: "the foundational work of Hawthorne et al. (2018), most music is highly structured and can be represented in a variety of forms" --- The cited source is hardly the first work to realize that music is highly structured and has a variety of forms. Please find a better reference.
- Line 318: "labels obtained from participants very familiar with music and piano repertoire." --- Consider adding information on the number and experience level of human annotators and the amount of data annotated per annotator.
- Lines 465f are superfluous.

---

### Author Response · Authors · 2024-11-20
**Response to all reviewers**

Some reviewers have asked questions about significance. We emphasize that our submission is a 'datasets and benchmarks' paper, and we have taken significant measures to detail and quantitatively analyze our methodology. As outlined in the ICLR call for papers, the conference explicitly includes ‘datasets and benchmarks’ as a relevant topic, along with ‘music’ and ‘audio’ as application areas.

As some reviewers have noted, we argue our dataset is significant, being two orders of magnitude larger than previous datasets of its kind. The distillation method we introduced enabled us to remove approximately 99% of the non-piano audio from our crawled corpus. In contrast, if we had used the audio classification method we compare against in our ablation study, extrapolating from Table 5, we estimate that nearly 10,000 hours of non-piano audio would have been included in the dataset. We imagine that our distillation method for audio classification and segmentation could have broad applications for other audio datasets: We are open-sourcing all of our data pipeline tools, as well as the new segmentation accuracy benchmark we propose.

Motivated by reviewer NngF’s feedback, we are including additional supplementary material from a generative experiment. We trained a large autoregressive transformer model on Aria-MIDI using a tokenizer similar to that used for MuseNet [1]. For comparison, we provide samples from an identical model trained on the GiantMIDI dataset. In both cases we used the MAESTRO dataset as a validation set, implementing early stopping in order to prevent overfitting in the case of GiantMIDI. We urge you to put on headphones and listen to the samples. Non-cherry-picked continuations (2000 tokens) of prompts spanning different genres can be heard here:

https://aria-midi.github.io/website/

We welcome any additional questions reviewers may have about this supplementary material.


[1] Payne, Christine. "MuseNet." OpenAI, 25 Apr. 2019, openai.com/blog/musenet

---

### Meta-Review · Area_Chair_p2ht · 2024-12-20

**Metareview:**

The paper presents a new dataset of MIDI piano recordings derived from audios scraped from YouTube videos. The impressive aspect of this work is the sizable amount of piano recordings in the dataset that is one or two order of magnitudes larger then currently existing datasets. The pipeline for creating the dataset uses existing methods comprising of an Automatic Music Transcription model, a language model to score candidate video descriptions and a segmentation into piano/non-piano segments.
The authors promise to make the pipeline publicly available, thought the details of the AMT method were not disclosed for review reasons. Moreover, in response review comment, the authors conducted a generative experiment, providing examples but without conducting an objective analysis of the quality. From my listening to the examples, the generative model trained on the new dataset seemed to be consistently more coherent and remained "in style" relative to an alternative method.

**Additional Comments On Reviewer Discussion:**

The main concern of the reviewers was the originality of the method, which is indeed based on existing methods. Since as the authors point out, ICLR solicits new datasets, and since the shear size of the new dataset and the effort to create it seem significant, my recommendation is not to take into account this as a major concern. In terms of other technical or more specific concerns, reviewer VtYa pointed out some potential issues with accuracy and ablation, to which the authors seem to have provided mostly adequate response. The response to reviewer NngF request to conduct a generative experiment was addressed, alas only qualitatively, providing listening examples.

---

### Decision · Program_Chairs · 2025-01-22

Accept (Poster)